# A Mixed-Perception Approach for Safe Human–Robot Collaboration in Industrial Automation

**DOI:** 10.3390/s20216347

**Published:** 2020-11-07

**Authors:** Fatemeh Mohammadi Amin, Maryam Rezayati, Hans Wernher van de Venn, Hossein Karimpour

**Affiliations:** 1Institute of Mechatronics System, Zurich University of Applied Science, 8400 Winterthur, Switzerland; mohm@zhaw.ch (F.M.A.); rzma@zhaw.ch (M.R.); 2Mechanical Engineering Department, University of Isfahan, Isfahan 81746-73441, Iran; h.karimpour@eng.ui.ac.ir

**Keywords:** safe physical human–robot collaboration, collision detection, human action recognition, artificial intelligence, industrial automation

## Abstract

Digital-enabled manufacturing systems require a high level of automation for fast and low-cost production but should also present flexibility and adaptiveness to varying and dynamic conditions in their environment, including the presence of human beings; however, this presence of workers in the shared workspace with robots decreases the productivity, as the robot is not aware about the human position and intention, which leads to concerns about human safety. This issue is addressed in this work by designing a reliable safety monitoring system for collaborative robots (cobots). The main idea here is to significantly enhance safety using a combination of recognition of human actions using visual perception and at the same time interpreting physical human–robot contact by tactile perception. Two datasets containing contact and vision data are collected by using different volunteers. The action recognition system classifies human actions using the skeleton representation of the latter when entering the shared workspace and the contact detection system distinguishes between intentional and incidental interactions if physical contact between human and cobot takes place. Two different deep learning networks are used for human action recognition and contact detection, which in combination, are expected to lead to the enhancement of human safety and an increase in the level of cobot perception about human intentions. The results show a promising path for future AI-driven solutions in safe and productive human–robot collaboration (HRC) in industrial automation.

## 1. Introduction

As the manufacturing industry evolves from rigid conventional procedures of production to a much more flexible and intelligent way of automation within the frame of the Industry 4.0 paradigm, human–robot collaboration (HRC) has gained rising attention [1,2]. To increase manufacturing flexibility, the present industrial need is to develop a new generation of robots that are able to interact with humans and support operators by leveraging tasks in terms of cognitive skills requirements [1]. Consequently, the robot becomes a companion or so-called collaborative robot (cobot) for flexible task accomplishment rather than a preprogrammed slave for repetitive, rigid automation. It is expected that cobots actively assist operators in performing complex tasks, with highest priority on human safety in cases humans and cobots need to physically cooperate and/or share their workspace [3]. This is problematic because the current settings of cobots do not provide an adequate perception of human presence in the shared workspace. Although there are some safety monitoring systems [4,5,6,7], they can only provide a real or virtual fence for the cobot to stop or slow down when an object, including a human being, enters the defined safety zone. However, this reduces productivity in two ways as follows:
It is not possible to differentiate between people and other objects that enter the workspace of the cobot. Therefore, the speed is always reduced regardless of the object.It is also not possible to differentiate whether an interaction with the robot should really take place or not; this also always forces a maximum reduction in speed.

This issue can only be tackled by implementing a cascaded, multi-objective safety system, which primarily recognizes human actions and detects human–robot contact [8] to percept human intention in order to avoid collisions. Therefore, the primary goal of this work is to conduct a step-change in safety for HRC in enhancing the perception of cobots by providing visual and tactile feedback to the robot from which it is able to interpret the human intention. The task is divided into two parts, human action recognition (HAR) using visual perception and contact type detection using tactile perception, which will be subsequently investigated. Finally, by combining these subsystems, it is considered to attain a more reliable and intelligent safety system, which takes advantage of considerably enhanced robot perceptional abilities.

### 1.1. Human Action Recognition (HAR)

Based on the existing safety regulation related to HRC applications and by inspiring from human perception and cognition ability in different situations, adding the visual perception to the cobot can enhance HRC performance. Nevertheless, the main challenge is how cobots are able to adapt to human behavior. HAR as part of visual perception plays a crucial role in overcoming this challenge and increasing productivity and safety. HAR can be used to allow the cobot keeping a safe distance with its human counterpart or the environment, ensuring an essential requirement for fulfilling safety in shared workspaces. Recent studies have been concentrated on visual and non-visual perception systems to recognize human actions [9]. One method amongst non-visual approaches consists of using wearable devices [10,11,12,13,14,15]. Nevertheless, applying this technology as a possible solution for an industrial situation seems at present neither feasible nor comfortable in industrial environments because of restrictions that it will impose on the operator’s movements. On the other hand, active vision-based systems are widely used in such applications for recognizing human gestures and actions. In general, vision-based HAR approaches consist of two main steps: proper human detection and action classification.

As alluded by recent researches, machine learning methods are essential in recognizing human actions and interpreting them. Some traditional machine learning methods such as support vector machine (SVM) [16,17,18,19], hidden Markov model (HMM) [20,21], neural networks [22], and Gaussian mixture models (GMM) [23,24] have been used for human action detection with a reported accuracy of about 70 to 90 percent. On the other hand, deep learning (DL) algorithms prevail as a new generation of machine learning algorithms with significant capabilities in discovering and learning complex underlying patterns from a large amount of data [25]. This algorithm provides a new approach to improve the recognition accuracy of human actions by using depth data provided by time-of-flight, depth, or stereo cameras, extracting human location and skeleton pose. DL researchers either use video stream data [26,27,28], RGB-D images [29,30,31], or 3D skeleton tracking and joints extraction [32,33,34,35] for classification of arbitrary actions. Among different types of deep networks, convolutional neural networks (CNN) stand for a popular approach, which can be represented as 2D or 3D network in action recognition but still needs a large set of labeled data for training and contains many layers. The first 3D-CNN for HAR has been introduced by [36,37,38] providing an average accuracy of 91 percent. Recent researches based on 3D-CNN techniques [39,40,41,42] have obtained a high performance on the KTH dataset [43] in comparison to 2D-CNN networks [44,45,46,47]. Yet, the maximum accuracy of this research is reported to be at 98.5 percent but is not capable of classifying in real-time. In addition, most of these articles mainly focus on action classification based on domestic scenarios, only few have an approach for industrial scenarios [19,48,49] and a restricted number works on unsupervised human activities in the presence of mobile robots [50,51]. Thus, there is a need to introduce a fast and more precise network for HAR in industrial applications, which can be presented as a new 3D network architecture by considering an outperforming result in action classification.

In this work, we use a deep learning approach for real-time human action recognition in an industrial automation scenario. A convolutional analysis is applied on RGB images of the scene in order to model the human motion over the frames by skeleton-based action recognition. The artificial-intelligence-based human action recognition algorithm provides the core part for distinguishing between collision and intentional contact.

### 1.2. Contact Type Detection

In more and more HRC applications, there is a need of having direct, physical collaboration between human and cobot, physical human–robot collaboration (pHRC) due to an unmatched degree of flexibility in the execution of various tasks. Indeed, when a cobot is performing its task, it should be aware of its contact with the human. In addition, from a cobot’s point of view, the type of this contact is not immediately obvious, due to the fact that the cobot cannot distinguish whether a human gets in contact with the robot incidentally or intentionally, when a collaborative task is executed. Therefore, it is important that the cobot needs to percept human contact with deeper understanding. Towards this goal, it is imperative firstly, to detect the human–robot contact and secondly, distinguish between intentional and incidental contacts, a process called collision detection. Some researchers propose sensor-less procedures for detecting a collision based on the robot dynamics model [52,53], but through momentum observers [52,54,55,56,57], using extended state observers [58], vibration analysis models [59], finite-time disturbance observers [56], energy observers [57], or joint velocity observers [60]. Among these methods, the momentum observer is the most common method of collision detection, because it provides better performance compared to the other methods, although the disadvantage is that it requires precise dynamic parameters of the robot [61]. For this reason, machine learning approaches such as artificial neural networks [62,63,64] and deep learning [65,66] have recently been applied for collision detection based on robot sensors’ stream data due to their performance in modeling the uncertain systems with lower analytics effort.

Deep neural networks are extremely effective in feature extraction and learning complex patterns [67]. Recurrent neural networks (RNN) such as long short-term memory network approaches (LSTM) are frequently used in research for processing time series and sequential data [68,69,70,71]. However, the main drawback of this network is the difficulty and time consumption for training in comparison to convolutional neural networks (CNN) [65]. Additionally, current researches showed that CNN has a great performance for image processing in real time situations [26,65,72,73,74], where the input data are much more complicated than 1D time series signals. As proposed in [65], a 1D-CNN, named CollisionNet, has a proper potential in detecting collision, although only incidental contacts have been considered. Moreover, depending on whether the contact is intentional or incidental, the cobot should provide an adequate response, which in every case, ensures the safety of the human operator. At this step, identifying the cobot link where the collision occurred is important information for anticipating proper robot reaction, which needs to be considered in contact perception [61].

With this background and considering the fact that contact properties´ patterns of incidental and intentional states are different according to the contacted link, we aim to use supervised learning, convolutional neural network, to have a model-free contact detection. Indeed, not only does the proposed system detect the contact, which in other papers [61,65,75,76] is named collision detection, it can also recognize the types of contact, incidental or intentional, provide information about a contacted link and consequently increase the robot awareness and perception about human intention during physical contact.

## 2. Material and Methods

### 2.1. Mixed Perception Terminology and Design

We hypothesize that combining two types of perception, visual and tactile, in a mixed perception approach can enhance the safety of human during collaborating with a robot by additional information to the robot’s perception spectrum. It is easy to imagine that a robot then would be able to see and feel a human in its immediate vicinity at the same time. Using visual perception, a robot can notice:
A human entering the shared workspace (Passing)A human observing its tasks when he/she is near to the robot and wants to supervise the robot task (Observation)A dangerous situation when the human is not in a proper situation to do collaboration or observation, which can threaten human safety (Dangerous Observation)A human interaction when the human is close to robot and doing the collaboration (Interaction).

However, it is difficult using a pure vision-based approach to distinguish between dangerous observation and interaction and to differentiate between incidental and intended contact types not only for a machine but also for a human. Therefore, at this stage, considering both types of perception, vision and haptics, is of significance. As indicated above, this approach is able to increase the safety and can be like a supervisory unit to the vision part as the latter can fail due to occlusion effects.

To support our hypothesis, we first choose the approach of developing two separate networks for human action and contact recognition, which meet the requirements for human–robot collaboration and real-time capability. These networks will be examined and discussed with regard to their appropriateness and their results. As a first step, we want to determine in this paper whether a logical correlation of the outputs of the two networks is theoretically able to provide a reasonable expansion of the perception spectrum of a robot for human–robot collaboration. We want to find out what the additional information content is and how it can specifically be used to further increase the safety and with that possibly also additional performance parameters of HRC solutions such as short cycle time, low downtime, high efficiency, and high productivity. The concrete merging of the two networks in a common application represents an additional stage of our investigations, which is not a subject of this work. The results of the present investigation, however, shall provide evidence that the use of AI in robotics is able to open up significant new possibilities and enables robots to achieve their operational objectives in close cooperation with humans. Enhanced perceptional abilities of robots are future key features to shift the existing technological limits and open up new fields of application in industry and beyond.

### 2.2. Robotic Platform

The accessible platform used throughout this project is a Franka Emika robot (Panda), recognized as a suitable collaborative robot in terms of agility and contact sensitivity. The key features of this robot will be summarized hereafter; it consists of two main parts, arm and hand. The arm has 7 revolute joints and precise torque sensors (13 bits resolution) at every joint, is driven by high efficiency brushless dc motors, and has the possibility to be controlled by external or internal torque controllers at a 1 kHz frequency. The hand is equipped with a gripper, which can securely grasp objects due to a force controller. Generally, the robot has a total weight of approximately 18 kg and can handle payloads up to 3 kg.

### 2.3. Camera Systems

The vision system is based on a multi-sensor approach using two Kinect V2 cameras for monitoring the environment to tackle the risk of occlusion. The Kinect V2 has a depth camera with resolution of 512 × 424 pixels with a field of view (FoV) of 70.6° × 60°, and the color camera has a resolution of 1920 × 1080 px with a FoV of 84.1° × 53.8°. Therefore, this sensor as one of the RGB-D cameras can be used for human body and skeleton detection.

### 2.4. Standard Robot Collision Detection

A common collision detection approach is defined as Equation (1) [61].
(1)$$cd\left(\mu \left(t\right)\right)=\{\begin{array}{c} TRUE,\;if\left|\mu \left(t\right)\right|>\epsilon_{\mu }\\ FALSE,\;if\left|\mu \left(t\right)\right|\leq \epsilon_{\mu } \end{array}$$
where *cd* is the collision detection output function, which maps the selected monitoring signal *μ*(*t*) such as external torque into a collision state, true or false. *ϵ_µ_* indicates a threshold parameter, which determines the sensitivity for detecting a collision.

### 2.5. Deep Learning Approach

A convolutional neural network (CNN) model performs classification in an end-to-end manner and learns data patterns automatically, which is different to the traditional approaches where the classification is done after feature extraction and selection [77]. In this paper, a combination of 3D-CNN for HAR and 1D-CNN for contact type detection has been utilized. The following subsections describe each network separately.

#### 2.5.1. Human Action Recognition Network

Since human actions can be interpreted by analyzing the sequence of human body movements involving arms and legs and placing them in a situational context, the consecutive skeleton images are used as inputs for our 3D-CNN network, which was successfully applied for real-time action recognition. In this section, the 3D-CNN, which is shown in Figure 1, classifies HAR to five states, namely: Passing, Observation, Dangerous Observation, Interaction, and Fail. These categories are based on the most feasible situations which may occur during human–robot collaboration:
Passing: a human operator occasionally needs to enter the robot’s workspace, which is specified due to the fix position of the robot but without any intention to actively intervene the task execution of the robot.Interaction: a human operator wants to actively intervene the robot’s task execution, which can be the case due to correct a Tool Center Point (TCP) path or to help the robot if it gets stuck.Observation: the robot is working on its own and a human operator is about to observe and check the working process from within the robot’s workspace.Dangerous Observation: the robot is working on its own and a human operator is about to observe the working process. Due to the proximity of exposed body parts (head and upper extremities) to the robot in the shared workspace, there is a high potential of life-threatening injury in case of a collision.Fail: one or all system cameras are not able to detect the human operator in the workspace due to occlusion by the robot itself or other artefacts in the working area.

The input layer has 4 dimensions, N_channel_ × N_image-height_ × N_image-width_ × N_frame_. The RGB image of Kinect V2 has a resolution of 1980 × 1080 pixels which is decreased to 320×180 for reducing the trainable parameters and network complexity. Therefore, N_channel_, N_image-height_, and N_image-width_ are 3, 180, and 320, respectively. N_frame_ indicates the total number of frames in the image sequence, which is 3 in this research.

As shown in Figure 1, the proposed CNN is composed of fifteen layers, consisting of four convolutional layers, four pooling layers, three fully connected layers followed by three dropout layers and a SoftMax layer for predicting actions. Convolutional layers utilized for feature extraction by applying filters and pooling layers are specifically used to reduce the dimensionality of the input. This layer performs based on the specific function, such as max pooling or average pooling, which extracts the maximum or medium value in a particular region. Fully connected layers are like a neural network for learning non-linear features as represented by the output of convolutional layers. In addition, dropout layers as a regularization layer try to remove overfitting in the network. Over 10 million parameters have to be trained for establishing a map to action recognition.

The input layer is followed by a convolution layer with 96 feature maps of size 7^3^. Subsequently, the output is fed to the rectified linear unit (ReLU) activation function. ReLU is the most suitable activation function for this work, as it is specifically designed for image processing, and it can keep the most important features of the input. In addition, it is easier to train and usually achieves better performance, which is significant for real-time applications. The next layer is a max-pooling layer with size and stride of 3. The filter size of the next convolutional layers decreases to 5^3^ and 3^3^, respectively, with stride 1 and zero padding. Then, max-pooling windows decline to 2^3^ with stride of 2. The output of the last pooling layer is flattened out for the fully connected layer input. The fully connected layers consist of 2024, 1024, 512 neurons, respectively. The last step is to use a SoftMax level for activity recognition.

#### 2.5.2. Contact Detection Network

For contact detection, a deep network, which is shown in Figure 2, is proposed. In this scheme, a 1D-CNN, which is a multi-layered architecture with each layer consisting of few one-dimensional convolution filters, is used. In this research, just two links of the robot which are more likely to be used as contact points during physical human–robot collaboration, considered which indeed does not influence the general approach used in this paper. Therefore, it includes one network for classification of 5 states, which were defined as:
No-Contact: no contact is detected within the specified sensitivity.Intentional_Link5: an intentional contact at robot link 5 is detected.Incidental_Link5: a collision at robot link 5 is detected.Intentional_Link6: an intentional contact at robot link 6 is detected.Incidental_Link6: a collision at robot link 6 is detected.

In this paper, the input vector represents a time series of robot data as
(2)$$x=\left[\begin{array}{cc} \begin{array}{cc} \tau_{J}^{0} & \tau_{ext}^{0}\\ \tau_{J}^{1} & \tau_{ext}^{1} \end{array} & \begin{array}{cc} q^{0} & {\dot{q}}^{0}\\ q^{1} & {\dot{q}}^{1} \end{array}\\ \begin{array}{cc} \vdots & \vdots \\ \tau_{J}^{W} & \tau_{ext}^{W} \end{array} & \begin{array}{cc} \vdots & \vdots \\ q^{W} & {\dot{q}}^{W} \end{array} \end{array}\right]$$
and
(3)$$\tau_{J}^{i}\;=\left[\tau_{j1}^{i}\;\tau_{j2}^{i}\;\tau_{j3}^{i}\;\tau_{j4}^{i}\;\tau_{j5}^{i}\;\tau_{j6}^{i}\;\tau_{j7}^{i}\right]$$
(4)$$\tau_{ext}^{i}=\left[\tau_{ext1}^{i}\;\tau_{ext2}^{i}\;\tau_{ext3}^{i}\;\tau_{ext4}^{i}\;\tau_{ext5}^{i}\;\tau_{ext6}^{i}\;\tau_{ext7}^{i}\right]$$
(5)$$q^{i}\;=\left[q_{1}^{i}\;q_{2}^{i}\;q_{3}^{i}\;q_{4}^{i}\;q_{5}^{i}\;q_{6}^{i}\;q_{7}^{i}\right]$$
(6)$$\dot{q^{i}}\;=\left[{\dot{q}}_{1}^{i}\;{\dot{q}}_{2}^{i}\;{\dot{q}}_{3}^{i}\;{\dot{q}}_{4}^{i}\;{\dot{q}}_{5}^{i}\;{\dot{q}}_{6}^{i}\;{\dot{q}}_{7}^{i}\right]$$
where *τ_J_, τ_ext_, q*, and $\dot{q}$ indicate joint torque, external torque, joint position, and joint velocity, respectively. W is the size of a window over the collected signals, which stores time-domain samples as an independent instance for training the proposed models. Hence, the input vector is W × 28, and in this research, by selecting 100, 200, and 300 samples for W, three different networks were trained to compare the influence of this parameter.

As shown in Figure 2, the designed CNN is composed of eleven layers. In the first layer of this model, the convolution process maps the data with 160 filters. The kernel size of this layer is optimally considered 5 to obtain a faster and sensitive enough human contact status; a parameter higher than 5 led to an insufficient network’s response, as it is more influenced by past data rather than near to present data. To normalize the data and avoid overfitting, especially due to the different maximum force patterns of every human, a batch normalization is used in the second layer. Furthermore, the size of all max pooling layers is chosen as 2, and ReLU function is considered as the activation function, due to reasons already mentioned before.

### 2.6. Data Collection

#### 2.6.1. Human Action Recognition

The HAR data are collected simultaneously from different views by two Kinect V2 cameras recording the scene of an operator moving next to a robot performing repetitive motions. The human skeleton is detected using the Kinect library based on the random forest decision method [78]. As the Kinect V2 library in Linux is not precise and does not project human skeleton in RGB images, the 3D joint position in depth coordination was extracted and converted to RGB coordinates as follows:(7)$$x_{rgb}=x_{d}\times \frac{PD_{xrgb}}{PD_{xd}}+\frac{C_{xrgb}\times PD_{xd}-C_{xd}\times PD_{xrgb}}{PD_{xrgb}\times PD_{xd}}$$
(8)$$y_{rgb}=y_{d}\times \frac{PD_{yrgb}}{PD_{yd}}+\frac{C_{yrgb}\times PD_{yd}-C_{yd}\times PD_{yrgb}}{PD_{yrgb}\times PD_{yd}}$$
where (*C_xrgb_, C_yrgb_*) and (*C_xd_, C_yd_*) are RGB and depth image centers, respectively. PD shows the number of pixels per degree for depth and RGB images, respectively equal to 7 × 7 and 22 × 20 [79,80]. Then, the RGB images, which are supplemented with the skeleton representation in each frame, are collected as a dataset. The sample rate by considering the required time for saving the images was 22 frames/second. Both cameras start collecting data once the human operator enters the environment, while it is assumed that the robot is stationary in a structured environment. The collected images are then sorted into 5 different categories and labeled accordingly based on the skeleton position and configuration and with respect to the fixed base position of the robot.

#### 2.6.2. Contact Detection

The data acquired at the robot joints during a predefined motion with a speed of 0.5 m/s were collected in three states, contact-free, during interaction with the operator, and collision-like contacts, at a sampling rate of 200 Hz (one sample per 5 ms). In this part, collecting collision-like contact data is challenging, as the dedicated operator induces the collision intentionally [65]. However, the collision can be considered to happen in a normal situation where the human is standing with no motion and the robot is performing its task, while the impact happens. Indeed, a data analysis shows that it can be clearly distinguished from object and intentional contacts and therefore can be used at least as similar samples of real collision data. Then, a frame of W-window with 200 ms latency passed through the entire data gathered, preparing it to be used as training data for the input layer of the designed network. Thanks to the default cartesian contact detection ability of the Panda robot, those contact data are used as a trigger to stop recording data after contact occurrence. Consequently, the last W-samples of each collision trial data is considered as input for training the network. For assuring comprehensiveness of the gathered data, each trial is repeated 10 times with different scenes, including touched links, direction of motion, line of collision with the human operator, and contact type (intentional or incidental). Additionally, each sample is labeled according to the mentioned sequence.

### 2.7. Training Hardware and API Setup

In the training of a network by using Graphic Processor Units (GPU), memory plays an important role in reducing the training time. In this research, a powerful computer with NVIDIA Quadro P5000 GPU, Intel Xeon W-2155 CPUs, and 64 GB of RAM is employed for modeling and training the CNN networks using the Keras library of TensorFlow. To enable CUDA and GPU-acceleration computing, a GPU version of TensorFlow is used, and in consequence, the training process is speeded up. The total runtime of the vision network trained with 30,000 image sequences was about 12 h for 150 epochs, while it was less than 5 min for training contact networks.

### 2.8. Real Time Interface

The real-time interface for collecting the dataset and implementing the trained network on the system was provided by Robotics Operating System (ROS) on Ubuntu 18.04 LTS. Figure 3 shows the hardware and software structure used in this work. Two computers execute the vision networks for each camera separately and publish the action states at the rate of 200 Hz on ROS. Furthermore, CDN and CDM are executed on another PC at the same rate, connected to the robot controller for receiving the robot torque, velocity, and position data of joints 5 and 6.

## 3. Results

In order to evaluate the performance of the proposed system, the following metrics are used. A first evaluation consists of an offline testing, for which the results are compared based on the key figures precision, recall, and accuracy, defined as follows:
(9)$$Precision=\frac{tp}{tp+fp}$$
(10)$$Recall=\frac{tp}{tp+fn}$$
(11)$$Accuracy=\frac{t_{p}+t_{n}}{t_{p}+f_{n}+t_{n}+f_{p}}$$
where *t_p_* is the amount of the predicted true positive samples, *t_n_* is the number of data points labeled as negative correctly, *f_p_* represents the amount of the predicted false positive samples, and *f_n_* is the count of predicted false positive classes.

The second evaluation is based on real-time testing; the tests have shown promising results in early trials. The YouTube video (https://www.youtube.com/watch?v=ED_wH9BFJck) gives an impression of the performance (due to safety reasons, the velocity of the robot is reduced to an amount, which is considered to be safe according to ISO 10218).

### 3.1. Dataset

Regarding the vision category, the dataset consisting of 33,050 images is divided into five classes, including Interaction, Observation, Passing, Fail, and Dangerous Observation, with Figure 4 representing the different possible actions of a human operator during robot operation. The contact detection dataset [81] with 1114 samples is subdivided into five classes, namely No-Contact, Intentional_Link5, Intentional_Link6, Incidental_Link5, Incidental_Link6, which determine the contact state on the last two links including their respective type, incidental or intentional.

### 3.2. Comparison between Networks

#### 3.2.1. Human Action Recognition

For optimizing efficiency in HAR, two different networks, 2D and 3D, were tested, the latter indicating a significant outcome in both real-time and off-line testing cases. These two networks are compared with respect to the results of 150 training epochs, in Table 1**.** The confusion matrix can be considered as a good measurement to deliver the overall performance in the multi-category classification systems. As it is shown in Table 2, each row of the table represents the actual label, and each column indicates the predicted labels, which can also show the number of failed predictions in every class. As shown in Table 1, both networks have promising result in classifying “Interaction”, “Passing”, and “Fail” states. However, these networks have lower, but sufficient, accuracy in classifying the “Dangerous Observation” category due to the lack of third dimensional (depth) information in the network input. By considering the confusion matrix shown in Table 2, it is obvious that the networks did not precisely distinguish between “Interaction”, “Observation”, and “Dangerous Observation” caused by the similarity of these three classes. With regard to the condition of the experimental setup where the location of cameras and robot base are fixed, the current approach has enough accuracy, but for a real industry case, we need to add a true 3D representation of the human skeleton and the robot arm in our network input.

#### 3.2.2. Contact Detection

To evaluate the influence of the size of the sampling window (w) on the precision of the trained networks, three different size dimensions of 100, 200, and 300 unity are selected, corresponding to 0.5, 1, 1.5 s of sampling period duration. Seventy percent of the dataset is selected for training and 30% for testing. Each network is trained with 300 epochs, and the results are shown in Table 3 and Table 4. Window size of 200 and 300 unities provide a good precision for identifying the contact status, in contrast to w = 100, which is not satisfactory. Furthermore, by comparing the result of the 200-window and 300-window networks, the 200-window network provides a better precision and recall.

#### 3.2.3. Mixed Perception Safety Monitoring

Every perception system designed separately to detect human intention according to Figure 5a,b is regarded as a preliminary condition for the mixed perception system shown in Figure 5c. As shown in Figure 5, for proper safety monitoring, the robot is programmed to categorize human safety into three levels—Safe, Caution, and Danger—with its respective color codes green/yellow, orange, and red. Safe level consist of two states, indicating whether the cobot has physical contact with human (yellow) or not (green). Considering only visual perception or only tactile perception in determining the safety level does not provide sufficient information compared to the mixed perception system. For instance, in green Safe state of mixed perception, the robot can have a higher speed and in consequence, increased productivity, while in the other perception systems, green Safe does not give this confidence to the robot to be faster; consequently, it should be more conservative about possible collisions. Thus, this higher information content can increase human safety and the robot’s productivity of pHRC systems. Already a simple logical composition of the results (Figure 5c) shows a significantly higher information content and thus a possible increase in safety and productivity in human–robot collaboration. However, it might be that the mixed perception approach will have multiple effects on the safety of HRC. Therefore, we will examine in detail the influence of the two subsystems on the overall performance and quality of the entire system at a later stage.

## 4. Discussion

Human–robot collaboration has recently gained a lot of interest and received many contributions on both theoretical and practical aspects, including sensor development [82], design of robust and adaptive controllers [83,84], learning robots force-sensitive manipulation skills [85], human interfaces [86,87], and similar. Besides, some companies attempt to introduce collaborative robots in order for HRC to become more suited to enter manufacturing applications and production lines. However, cobots available on the market have limited payload/speed capacities because of safety concerns, which limits HRC application to some light tasks with very limited productivity. On the other hand, according to the norms for HRC operations [88], it is not essential to observe a strict design or to limit the power of operations if human safety can be ensured in all its aspects. In this regard, an intelligent safety system as the mixed perception approach has been proposed in this research to detect hazardous situations to take care of the human safety from entering the shared workspace to physical interaction in order to jointly accomplish a task by taking advantage of visual and tactile perceptions. Visual perception detects human actions in the shared workspace. Meanwhile, tactile perception identifies human–robot contacts. One relevant piece of research in human action recognition focusing on industrial assembly application is mentioned in [88]. By taking advantage of RGB image and 3D-CNN network, the authors of the mentioned paper classified human action during assembly and achieved 82% accuracy [89], while our visual perception system shows higher accuracy of 99.7% by adding a human skeleton to the RGB series as the network input. Although our skeleton detection using Kinect library can be slightly affected by lighting conditions, it detects the human skeleton in near 30 FPS, which is essential for fast human detection in real-time HRC applications [90]. Indeed, using deep learning approaches such as OpenPose [91] and AlphaPose [92] can omit lightening problems [93]. However, their respective detection rates are 22 [91] and 23 FPS [92], which needs more researches to be faster and applicable in safety monitoring systems. Besides, among contact detection approaches in the state of the art, there are two similar works investigating collision detection using 1D-CNN. The authors of [94] compared both approaches, CollisionNet [65] and FMA [94], where the accuracy was 88% and 90%, respectively, featuring a detection delay of 200ms [94]. While our procedure in tactile perception (what is called collision detection in the state-of-the-art literature [61,65,75,76]) reached 99% with 80ms detection delay. For detecting contact type and robot joint, the accuracy was higher than 89% up to 96%, which in turn, needs more research to achieve a higher accuracy.

In this study, combining the result of both abovementioned intelligent systems is presented using a safety perception spectrum to examine the potential of the mixed perception approach in safety monitoring of collaborative workspaces. The result shows that even with a simple combination of both systems, the performance of safety monitoring can be improved as each system separately does not have enough perception of the collaborative workspace. Furthermore, this research suggests that the different forms of collaboration, such as coexistence, cooperation, etc., with their different safety requirements can be reduced to a single scenario using mixed perception as the robot would be able to “see” humans and “percept” external contacts.

As a result of this safety scenario, the robot reacts by being able to detect human intention, determining human safety level, and thus ensuring safety in all work situations. Another advantage of the proposed system is that the robot would be smart enough to take care about safety norms depending on the conditions and, consequently, could operate at an optimum speed during HRC applications. In other words, current safety requirements in most cases stop or drastically slow down the robot when a human enters a shared workspace. However, with the proposed safety system, based on the robots’ awareness using the presented mixed perception approach, it is possible to implement a reasonable trade-off between safety and productivity, which will be discussed in more detail in our future research.

In this research, there are two limitations concerning data collection: the collision occurred intentionally, and we did not gather data when the human and/or the robot move at high speed, which can be extremely dangerous for the human operator. As can be proved, the speed of the robot has an insignificant influence on the result, since the model has learned the dynamics of the robot in the presence or absence of human contact with normalized input data. On the other hand, if the human operator wants to grab the robot at high speed with the intention of working with it, this could be classified as a collision by the model due to its clear difference between contactless and intentional data patterns. However, this only increases the false positive error of the collision class (i.e., this would then be mistakenly perceived as a collision by the robot), which does not represent a safety problem in this case.

In addition, the current work focuses on a structured environment with fixed cameras and a stationary robot base position, which has yet to be generalized for an unstructured environment. In principle; however, this does not restrict the generality of this approach, since for cobots, only the corresponding position of the robot base has to be determined for the proximity detection to a human operator. In our ongoing work, we are trying to use some methods to tackle these problems. Moreover, with the current software and hardware, a sampling rate of HAR and contact detection networks are 30 Hz and 200Hz, respectively, while for the mixed perception system, there is a need for synchronization of the result of both systems.

## 5. Conclusions

The efficiency of safety and productivity of cobots in HRC can be improved if cobots are able to easily recognize complex human actions and can differentiate between multitude contact types. In this paper, a safety system using a mixed perception is proposed to improve the productivity and safety in HRC applications by making the cobot aware of human actions (visual perception), with the ability to distinguish between intentional and incidental contact (tactile perception). The vision perception system is based on a 3D-CNN algorithm for human action recognition, which unlike the latest HAR methods, was able to achieve 99.7% accuracy in an HRC scenario. The HAR system is intended to detect human action once the latter enters the workspace and only in case of hazardous situations, the robot would adapt its speed or stop accordingly, which can lead to higher productivity. On the other hand, the tactile perception, by focusing on the contact between robot and human, can decide about the final situation during pHRC. The contact detection system, by taking advantage of the contact signal patterns and 1D-CNN network, was able to distinguish between the incidental and intentional contact and recognize the impacted cobot’s link. According to the experimental result, with respect to traditional and new methods, our proposed model is obtained the highest accuracy of 96% in tactile perception.

Yet, based on our experimental results, visual and tactile perceptions are not sufficient enough separately for intrinsically safe robotic applications, since each system exhibits some lack of information, which may cause less productivity and safety. By considering this fact, the mixed perception, by taking advantage of both visual and tactile perception, can enhance productivity and safety. Although a simple safety perception spectrum of the mixed perception is proposed, which needs more research to enhance its intelligence, it shows higher resolution in compared to each single perception system.

As future work for our system, we will extend our research regarding to multi-contact and multi-person detection, which is highly beneficial for the latest Industry 4.0 safety considerations.

## Figures and Tables

**Figure 1 sensors-20-06347-f001:**
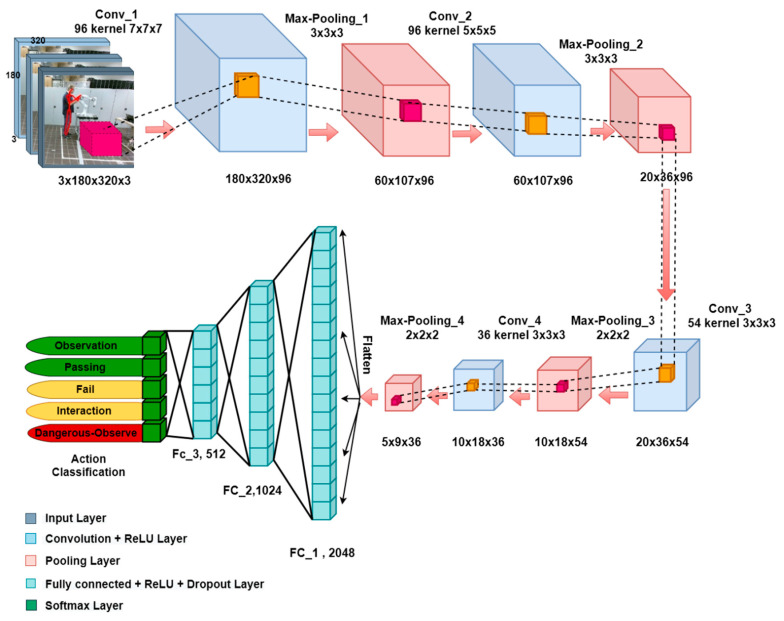
Three-dimensional convolutional neural networks (CNN) for human action recognition.

**Figure 2 sensors-20-06347-f002:**
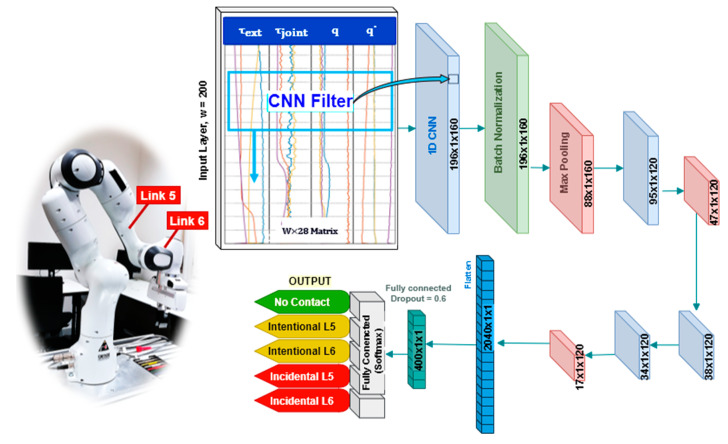
Contact detection network diagram.

**Figure 3 sensors-20-06347-f003:**
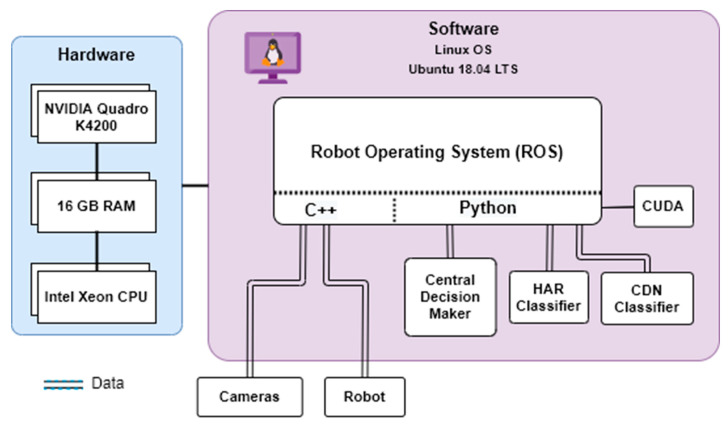
Real-time interface of complex system.

**Figure 4 sensors-20-06347-f004:**
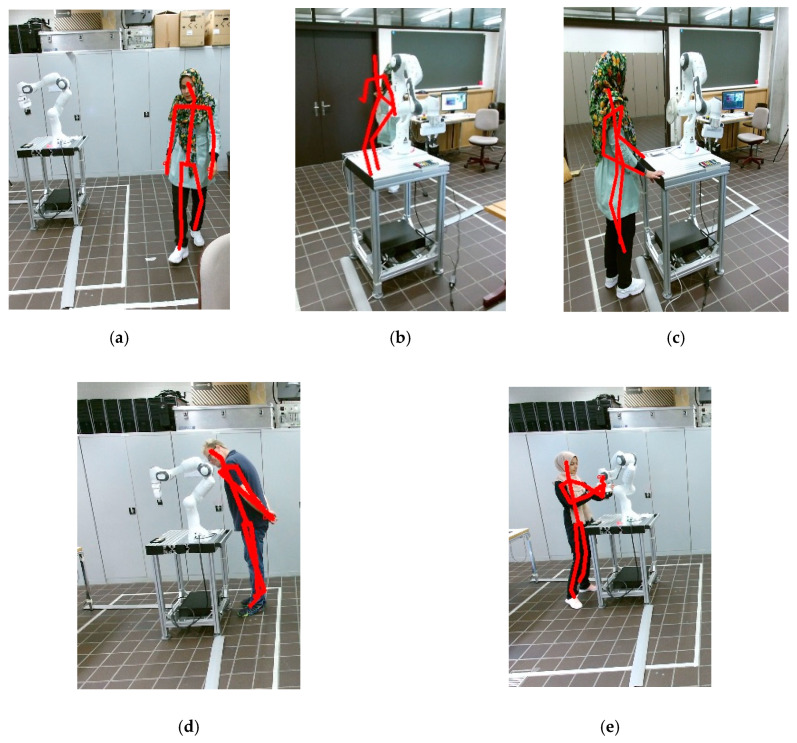
Type of human actions: (**a**) Passing: operator is just passing by, without paying attention to the robot. (**b**) Fail: blind spots or occlusion of the visual field may happen for a camera, in this situation the second camera takes over detection. (**c**) Observation: operator enters the working zone, without any interaction intention and stands next to the robot. (**d**) Dangerous Observation: operator proximity is too close, especially his head is at danger of collision with the robot. (**e**) Interaction: operator enters the working zone and prepares to work with the robot.

**Figure 5 sensors-20-06347-f005:**
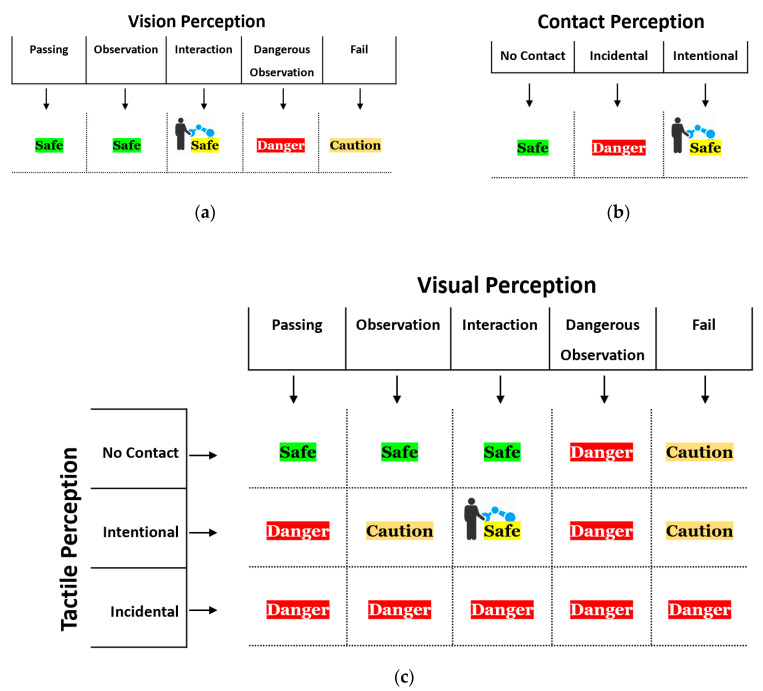
Safety perception spectrum in (**a**) visual perception, (**b**) contact perception, (**c**) mixed perception safety systems.

**Table 1 sensors-20-06347-t001:** Precision and recall of two trained networks for human action recognition.

Network	2D		3D	
	Precision	Recall	Precision	Recall
Observation	0.99	0.99	1.00	1.00
Interaction	1.00	1.00	1.00	1.00
Passing	1.00	1.00	1.00	1.00
Fail	1.00	1.00	1.00	1.00
Dangerous Observation	0.98	0.96	0.98	0.99
Accuracy	0.9956		0.9972	

**Table 2 sensors-20-06347-t002:** Confusion Matrix for different classes in HRC.

	Network	2D					3D				
		Observation	Interaction	Passing	Fail	Dangerous Observation	Observation	Interaction	Passing	Fail	Dangerous Observation
True Labels	Observation	3696	7	2	0	5	3751	6	2	1	7
	Interaction	13	4130	0	0	1	8	4030	0	0	0
	Passing	2	0	1145	0	0	1	0	1160	0	0
	Fail	0	0	0	593	0	0	0	0	588	0
	Dangerous Observation	12	1	0	0	313	2	0	0	0	359

**Table 3 sensors-20-06347-t003:** Precision and recall of trained networks for contact detection with different window size.

w	100	200	300	100	200	300
	Precision			Recall		
No-Contact	0.94	0.99	0.98	0.94	1.00	1.00
Intentional_Link5	0.74	0.91	0.89	0.84	0.91	0.84
Intentional_Link6	0.68	0.97	0.91	0.64	0.90	0.91
Incidental_Link5	0.61	0.89	0.83	0.61	0.93	0.89
Incidental_Link6	0.69	0.96	0.96	0.57	0.96	0.93
Accuracy	0.78	0.96	0.93			

**Table 4 sensors-20-06347-t004:** Confusion matrix of trained networks for contact detection with different window size.

	Window Size	100					200					300				
		No-Contact	Intentional_Link5	Intentional_Link6	Incidental_Link5	Incidental_Link6on	No-Contact	Intentional_Link5	Intentional_Link6	Incidental_Link5	Incidental_Link6	No-Contact	Intentional_Link5	Intentional_Link6	Incidental_Link5	Incidental_Link6
True Labels	No-Contact	166	0	9	0	1	242	0	3	0	1	167	0	3	0	0
	Intentional_Link5	0	86	12	19	0	0	93	4	4	1	0	86	5	5	1
	Intentional_Link6	8	1	59	2	17	0	3	83	0	0	0	5	84	0	3
	Incidental_Link5	0	15	1	33	5	0	6	0	50	0	0	10	0	48	0
	Incidental_Link6	3	0	11	0	31	0	0	2	0	52	0	1	0	1	50

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
