# Peer review of "A Mixed-Perception Approach for Safe Human–Robot Collaboration in Industrial Automation"

_sensors, 2020, doi:10.3390/s20216347_

Round 1

Reviewer 1 Report

This is an interesting article about a mixed perception approach for real time human-robot collaboration. The described problem is very important in the context of safety in smart manufacturing, as human behavior can be unpredictable.

The main part of the work is described in detail, but there are some gaps which require improvement.

Comments about methodology:

  • the initial assumptions should be described precisely, including the speed of the operator and robot, as this is the main factor in the risk of a collision
  • Why only two joints ? Which are they exactly?
  • the combination of ARN and CDN should be described in more detail. Why two levels of contacts (intentional, incidental) in Fig. 3 when the levels of danger are the same? How to distinguish “intentional” and “incidental” contact?
  • the discussion of the results (point 4.) is too general (Instructions for Authors: “Authors should discuss the results and how they can be interpreted in perspective of previous studies and of the working hypotheses. The findings and their implications should be discussed in the broadest context possible. Future research directions may also be highlighted.”)
  • The video shows that the operator and robot are moving rather slowly. What will happen if they are moving with greater speed (e.g. 2m/s)? How big will the delay of this system be and how this can change the results?

Some parts of the paper require clarification and edition, including:

Fehler! Verweisquelle konnte nicht gefunden werden.,

Fig. 1 – The “Max-Pooling_3” is twice

Line 147 - As shown in Figure 3 … - probably was supposed to be Fig. 1

Fig. 3 – Contact classifier is confusing as L5 shows up three times and L6 only once

Line 256: “Accuracy calculation follows later” … but the formula is missing.

The “Confusion matrix” in Figure 6 and 7 is somewhat confusing and additional description is needed

Other comments:

The video shows that sometimes there is a lag between the human and the skeleton representation of the body. How does this affect the operation of the system?

Reviewer 2 Report

In this paper authors have proposed a safe human-robot collaboration detection method in industrial automation, as indicated in the title. The safe human-robot collaboration becomes an important topic. However, lots of similar research have been presented. Authors should emphasize the innovation of this research, especially in theory and method part. The results are also needed to be compared with the more other methods presented before. Otherwise, it may make this research lack of value. In addition, this text is not well-organized for readers to read smoothly. Authors should consider how to improve the logic and readability.

  1. The article lacks a general description of the so-called Perception Approach, especially the process of the system. Readers hope to understand more intuitively how this perception method completes perception step by step from image input, if this research is describing a perception system rather than studying a certain technical detail such as Human Action Recognition Network or Contact Detection Network. Therefore, the structure of the Section 2, Material and Methods, should be checked and modified. Authors should explain the logic of the perception approach and highlight the main contribution of their work. As for the devices, hardware and software, they are necessary but replaceable theoretically, thus not the most important content, which should not occupy too much space.

  1. There are little comparison of essential parts in this work with the other existed similar or related methods, which causes that readers can neither identify whether the method used in this article is different from the other methods in principle, nor make sure that the methods used in this article is more suitable than the others. Take the 3D CNN Human Action Recognition network for example. In addition to some Human Action Recognition methods mentioned in the introduction, there are some other general recognition methods such as OpenPose (by Zhe Cao et al.), AlphaPose (by Hao-Shu Fang et al.) and especially 3D Convolutional Neural Networks for Human Action Recognition (by Shuiwang Ji et al.) can be used as comparison.

  1. The experiment of the article seems insufficient. The accuracy of the two networks have been tested separately, but the performance of the entire system is not. In addition, as mentioned above, some comparisons with the solutions that have been proposed are missing, which may lead to confusion about the results for readers.

  1. line 54-56: “On the other hand, … large changes in lighting conditions.”: Authors claims that active vision-based systems will be influenced by lighting conditions, but there are no related contents in the subsequent instructions, especially when the Deep Learning method they used doesn’t show the advantage to solve this shortage obviously.

  1. line 86-89: “For this reason, … their fast response and low computational cost.”: I have to say I have a little doubt here about how fast the response is and how low the calculation cost is. This conclusion can be drawn from experience, experiments or the other articles, and should not be too subjective at least in expression.

  1. line 132-134: “Since human actions can be, … applied for real-time action recognition.”: It seems that there are no descriptions about how to get the consecutive skeleton images. Are they generated by the camera system automatically, or some other algorithms are required?

  1. line 141:”Nimage-width × Nimage-height × Nchannel × Nframe ”: It would be better that the order here is consistent with the one in the figure above so that reader can match with each number more easily.

  1. line 166-168:”Link5”and”Link6”: What do the two numbers ,5 and 6, stand for? Maybe they are not important but would make readers confused.

  1. line 267 Figure 5: These five situations have been mentioned before but are introduced here. Maybe it is better to introduce the classification basis earlier.

  1. Authors should carefully check the layout of the article, including the charts, figures, formulas and hyperlinks. The English is straightforward, but I still recommend the double check.

Reviewer 3 Report

This paper proposes a mixed perception approach for real-time safe human-robot collaboration based on the combination of human action and contact type detection systems. The authors used two different deep learning networks for human action recognition and contact detection which can lead to the enhancement of human safety and an increase of the level of robot awareness about human intentions. The intent is good for robotic applications but there are a few concerns about the methodology and design of their experiments. Also there are several writing issues like obvious typos and layout problems, which the authors should carefully check before submitting the manuscript.

1. Can the authors clarify how the sampling is representative? How do they distinguish between intentional and incidental interactions when a physical contact between human and robot takes place? In fact, it would be difficult to collect incidental interaction data, since the volunteers essentially knew what they are doing during the tests.

2. The authors need to classify the five states of HAR, namely, passing, observation, dangerous observation, interaction, and fail. There is no explanation of the specific meaning of each state in the article.

3. Similarly, for the contact detection network, how do they classify the five states of the contact status? At the same time, can they explain why only link 5 and link 6 are considered in their model? Why are other links excluded? I don’t think this is reasonable.

4. The authors presented their network structure in Figures 1 and 2. However, can they explain the meaning of each layer of the networks?

Some other issues related to presentation:

- In Eq. (1), what is the specific monitoring signal you selected in the experiment?

- On line 140, “the input layer has 4 dimensions, Nimage-width × Nimage-height × Nchannel × Nframe,” which seems inconsistent with Figure 1.

- Many paragraphs do not have the first line indentation. Also, there are problems with the layout on pages 10 and 11. 

- On lines 185, 218, 240, there are very obvious typos that contain non-English text.

- The conclusion section should not just describe the basic steps for their methodology. They should also show the main results, observations and conclusions. They may also point out the potential future research topics.

Round 2

Reviewer 1 Report

The paper is now more readable and suitable for publication.

In next experiments, I suggest using a mannequin, not your own body.

Author Response

Dear reviewer,
Thank you for your many good tips and suggestions for improvement, which have made our paper really much better. We also take your last suggestion very seriously and will use a mannequin in our next experiments.
It's been a while, but please check out this video: https://www.youtube.com/watch?v=dnUwqngH0bM In the past people were obviously much more careless than they are today (by the way, Prof. Haddadin, shown here during his doctoral thesis, was quoted several times in this paper).  

Reviewer 2 Report

Well impoved.

One issue still need to be addressed.

In  the discussion section, references should be included in the newly added comparison. 

Author Response

Dear reviewer,

Thank you for your helpful tips and suggestions for improvement, which have made our paper really much better.
Please take the amendments according to your proposal from the attached PDF (additional literature sources marked in yellow).

Reviewer 3 Report

The revision makes improvements compared with the previous version, but there are still a few clarifications that the authors need to address:

(1) For deep learning networks, the label division is not clear.

(2) The specific signal collection is mostly to collect some data that the robot runs normally and has good human-machine interaction. It is difficult to collect abnormal data mainly for safety reasons. Can they comment on how this affects their model and their results?

Author Response

Dear Reviewer,

We are very grateful for your critical reading, valuable input and suggestions. It helped us a lot to further develop our paper and to improve it significantly. We hope that we were able to implement your suggestions well.

Please find our notes and the updated full text attached.

For the authors,

Wernher van de Venn
